# Extracellular Vesicles in Human Milk

**DOI:** 10.3390/ph14101050

**Published:** 2021-10-15

**Authors:** Yong Hu, Johannes Thaler, Rienk Nieuwland

**Affiliations:** 1Laboratory of Experimental Clinical Chemistry and Vesicle Observation Center, Amsterdam University Medical Center, University of Amsterdam, Meibergdreef 9, 1105 AZ Amsterdam, The Netherlands; y.hu@amsterdamumc.nl; 2Biomedical Engineering & Physics, Amsterdam University Medical Center, University of Amsterdam, Meibergdreef 9, 1105 AZ Amsterdam, The Netherlands; 3Clinical Division of Haematology and Haemostaseology, Department of Medicine I, Medical University of Vienna, Währinger Gürtel 18–20, 1090 Vienna, Austria; johannes.thaler@meduniwien.ac.at

**Keywords:** breast feeding, development, extracellular vesicles, infant, immune, inflammatory intestinal disease, milk

## Abstract

Milk supports the growth and development of infants. An increasing number of mostly recent studies have demonstrated that milk contains a hitherto undescribed component called extracellular vesicles (EVs). This presents questions regarding why milk contains EVs and what their function is. Recently, we showed that EVs in human milk expose tissue factor, the protein that triggers coagulation or blood clotting, and that milk-derived EVs promote coagulation. Because bovine milk, which also contains EVs, completely lacks this coagulant activity, important differences are present in the biological functions of human milk-derived EVs between species. In this review, we will summarize the current knowledge regarding the presence and biochemical composition of milk EVs, their function(s) and potential clinical applications such as in probiotics, and the unique problems that milk EVs encounter in vivo, including survival of the gastrointestinal conditions encountered in the newborn. The main focus of this review will be human milk-derived EVs, but when available, we will also include information regarding non-human milk for comparison.

## 1. Introduction

Milk is produced by mammary glands and is the “natural food” of infants [1]. Milk contains such nutrients as carbohydrates, lipids, minerals, proteins, and vitamins [2,3,4]. The composition of milk differs between species; for example, human milk is relatively rich in lactose compared to milk from cows, goats and sheep, whereas milk from cows, goats and sheep contains more protein than human milk [5].

Breast feeding is associated with beneficial effects for the infant. Infants fed with human breast milk have less intestinal complications than infants fed with formula [6]. However, milk may have more beneficial effects. A famous example dates back to the 1930s, when Alphons Sole, an Austrian pediatrician, reported that external bleeding of boys suffering from haemophilia, a hereditary bleeding disorder, is effectively stopped by pressing human milk-soaked tamponades on the bleeding spots [7,8]. At that time, the bioactive component in milk responsible for this effect was not identified.

The question is, which beneficial effects of milk can be attributed to which components?

An increasing number of publications suggests that at least some of these beneficial effects can be attributed to the presence of extracellular vesicles (EVs). “Extracellular vesicles” is an umbrella term for all phospholipid bilayer-enclosed particles that are released by cells into their environment. The term “extracellular vesicles” includes exosomes and microvesicles or microparticles, also known as ectosomes. Body fluids concurrently contain different types of EVs, but since there are no unique biochemical and/or physical properties known yet to distinguish one type of EV from the other, we prefer to use the term “extracellular vesicles”. Consequently, even when, for example, the term “exosomes” is used in a manuscript, we will use the term “extracellular vesicles” to circumvent confusion and misinterpretation. All body fluids, including amniotic fluid, blood, saliva, and urine, contain EVs under physiological as well as pathological conditions [9,10,11,12]. The presence of EVs in human milk was first shown by electron microscopy in 1980s [13], and since then, the presence of EVs has been demonstrated in milk from camels, cows, goats, and pigs [14,15,16,17]. In contrast, EVs are absent in formula milk [18].

The International Society for Extracellular Vesicles (ISEV) introduced the term “extracellular vesicles” and has worked to actively improve the rigor, standardization and credibility of EV research by publishing the *Minimal Information for Studies on Extracellular Vesicles* and accompanying checklists [19]. Since 2019, the ISEV has maintained a Rigor and Standardization Subcommittee (www.isev.org), and recently, an editorial was published in which ongoing efforts were summarized with the overall goal of improving the quality of EV research [20].

In the following section, we will summarize the thus far reported physical properties and cellular origin of EVs in milk.

## 2. Physical Properties and Cellular Origin of Extracellular Vesicles in Milk

The reported size range of EVs in human and bovine milk are comparable, as shown in Table 1. EVs in human milk are reported to range in diameter from 50 to 265 nm [21,22,23], and in bovine milk from 44 to 200 nm [24,25]. The absolute concentration of EVs in milk is unknown, because until now, only the total concentration of “particles”, i.e., including EVs plus non-EV particles, has been measured in cell-free fractions of human [26,27] and bovine milk [27,28].

The cellular origin of milk EVs is mostly unexplored. Human milk contains EVs exposing epithelial cell adhesion molecule (EpCAM), and such EVs likely originate from the mammary epithelium [33]. Bovine milk also contains such EVs [34]. Recent publications showed that milk contains immune cells [35], and thus, one would expect to observe immune cell-derived EVs. However, in our hands, using a sensitive flow cytometry capable of detecting single EVs with a diameter >160 nm, we were hardly able to detect EVs from leukocytes, monocytes, B-cells, and T-cells. Platelet-derived EVs were also not detectable, indicating that both immune cell-derived and blood cell-derived EVs are scarce or absent [33]. Whether these EVs are indeed hardly present, or whether most are still undetected due to their small size and the limited number of exposed antigens one needs to label and identify such (single) EVs, awaits further studies. Milk also contains bacteria [36,37,38], and bacterial EVs may present a fraction of EVs present in milk [38].

The mammary epithelium differentiates during lactation [39,40]. The changes in histology are reflected by changes in the composition of milk, which shifts from colostrum to mature milk [41]. It has been reported that colostrum EVs are superior compared to EVs from mature milk with regards to protection against intestinal inflammation due to the higher concentrations of immune- and growth-related proteins associated with colostrum EVs [42,43].

## 3. Isolation of Extracellular Vesicles from Milk

To gain insight into the functions of EVs present in milk, it is essential to isolate EVs to study their biochemical composition and/or study their function(s) directly. The current isolation methods that have been used to isolate milk EVs were adopted from those developed to isolate EVs from plasma or conditioned cultured medium. The 125 articles that were retrieved on PubMed (using the terms “milk and (extracellular vesicles or exosomes or microvesicles)”) showed that milk EVs are isolated by commonly used EV isolation methods as (ultra)centrifugation, density gradient centrifugation, commercial precipitation kits, and size exclusion chromatography (SEC). These methods have been extensively described elsewhere, including their advantages and disadvantages, but with regard to milk, a few additional points should be discussed and taken into account due to the complex composition of milk, including the presence of milk fat globules and casein micelles, as shown in Figure 1.

### 3.1. Removal of Cells and Cell Fragments

In most studies, raw milk is centrifuged at approximately 2000× *g* to remove cells and cell fragments. In none of these studies, however, has the efficacy of this procedure been confirmed by measuring the concentration of remaining cells and/or cell fragments after centrifugation. Because residual cells and cell fragments will co-isolate with all current isolation procedures applied to EVs, this is a potential confounder in most, if not all, studies published thus far.

### 3.2. Removal of Milk Fat Globules

During the centrifugation step to remove cells and cell fragments, the milk fat globules will start to float and form a layer. Skimming after centrifugation removes most milk fat globules, but neither the efficacy of skimming nor the presence of residual milk fat globules has been quantified. In some studies, as a second step, filtration, is applied to remove the remaining milk fat globules, but neither the efficacy of filtration nor the possible loss of EVs has been quantified thus far. Because size ranges of milk fat globules and EVs overlap, fractions of “isolated EVs” from milk are likely to be contaminated by milk fat globules. Their presence can be demonstrated by measuring the presence of milk fat globule–epidermal growth factor–factor 8 (MFG-E8). From the results of the reported protein composition of isolated milk-derived EVs, shown in Section 4.1. *Proteins* -, it is clear that “isolated EVs” do contain MFG-E8 and, thus, are contaminated with milk fat globules.

### 3.3. Removal of Casein Micelles

Another major hurdle is the presence of casein. Casein is the most common milk protein, accounting for up to 80% of total cow milk protein [44] and 35% of total human milk protein [28,45]. Casein forms spherical colloidal aggregates which are called casein micelles, and which range in size from 20 nm to 600 nm [31], and thus also overlap in size with EVs. The diameter of casein micelles is temperature dependent, and, for example, the casein micelles in human milk have an average diameter of 100 nm at 37 °C but 570 nm at 4 °C [46].

Multiple methods have been reported to remove casein (micelles) from milk, including centrifugation, acid precipitation, calcium ion chelation, and chymosin treatment. To which extent these procedures also affect the presence and (functional) integrity of EVs present in milk, however, is largely unexplored.

Centrifugation at 20,000–75,000× *g* removes a fraction of casein micelles. At higher g forces, e.g., at 340,000× *g*, casein micelles will crosslink and form a solid gel, which can be removed. However, even at this high centrifugation force, still not all casein can be removed [27,47], and such procedures may result in EV loss due to cross-linking of casein micelles and EVs [48]. Acid precipitation may damage EVs [49]. Since the casein micelle structure is calcium ion-dependent, chelation of calcium ions induces dissociation of casein micelles [50]. At high concentrations of ethylenediaminetetraacetic acid (EDTA), i.e., up to 50 mmol/L, casein micelles dissociate [51]. However, at such high concentrations of EDTA, EVs may also dissociate (R. Nieuwland, personal communication). Finally, chymosin hydrolyzes and degrades casein, and casein is not detectable in EVs isolated by ultracentrifugation after chymosin treatment [48]. Taken together, the methods to remove casein are diverse, but to which extent such methods affect the presence and functional activities of EVs in milk needs further investigation.

## 4. The Biochemical Composition of Extracellular Vesicles: Proteins and RNA

As explained in the previous section, the isolation of EVs is essential to gain insight into their biochemical composition and thus also their function(s). At present, EVs are widely considered to be vehicles supporting intercellular communication by delivering biologically active cargoes, e.g., proteins and RNAs, from the parent cell to recipient cells [52]. An overview of the composition of isolated EVs from human and non-human origin are shown for proteins (Table 2 and Table 3) and RNA (Table 4 and Table 5).

### 4.1. Proteins

Several databases contain reported proteomes of milk EV proteins, including ExoCarta (http://exocarta.org/index.html) accessed on 20 March 2021, and Vesiclepedia (http://microvesicles.org/index.html) accessed on 20 March 2021. Most of the identified surface proteins are involved in peptide transport, membrane biosynthesis, and proteolysis, whereas the identified cytosolic proteins are involved in functional cell signaling pathways [53]. Several proteomes have identified tetraspannins, such as CD9, CD63 and CD81, which are claimed to be specific EV markers [27] (Table 2). Because transmembrane proteins require a phospholipid bilayer membrane, which is provided only by EVs in milk and not by casein micelles or milk fat globules, the presence of tetraspannins confirms that indeed EVs were isolated and studied, although it does not rule out the presence of non-EV particles. Of note, tissue factor, which is abundantly present in human milk and exclusively associated with EVs, is lacking in the published proteomes, illustrating that the current databases are certainly still incomplete [54]. Additionally, the fact that there are considerable differences in the reported protein composition of milk-derived EVs, even within the milk of one species, indicates that validation is essential. At present, there are also no studies yet that demonstrate or confirm the functions of the identified proteins in milk-derived isolated EVs.

Recently, EVs from human plasma and serum were shown to be covered by a set of proteins. Such proteins, called “corona proteins”, may be involved in various functions [55], and it seems reasonable to assume that also EVs that are present in other body fluids may be covered with such corona proteins, although the composition of such a corona may be body fluid- and perhaps disease (state)-dependent.

**Table 2 pharmaceuticals-14-01050-t002:** Main proteins identified in isolated extracellular vesicles from human milk.

Isolation Procedure	Proteins	Detection	Study
Ultracentrifugation	CXCL5, SOD1, PRSS8, EPHA2, MIA, FR-alpha, MET, CD69, PDGF subunit B, CCL19	PEA	[56]
Ultracentrifugation	MHC class II, CD81, MUC1, HSPs, MFG-E8	FCM, WB, MS	[21]
Ultracentrifugation, size exclusive chromatography	TF, CD9, CD63	WB	[33]
Density gradient centrifugation	Colostrum: LCP1, IGHG1, IGKV3-20, LAMP1, PRDX1 Mature milk: SLC1A5, RPL10, Tetraspanin, GMPPA	LC-MS/MS	[57]
Density gradient centrifugation	CD9, CD63, MHC-class II, FLOT-1	LC-MS/MS, WB	[58]
Density gradient centrifugation	CD36, CD63, MUC1	FCM	[59]

**Table 3 pharmaceuticals-14-01050-t003:** Main proteins identified in isolated extracellular vesicles from livestock milk.

Milk source	Isolation Procedure	Protein	Detection	Study
Bovine	Ultracentrifugation	CD63, HSP70, CD9, CD81	WB	[60]
Bovine	Ultracentrifugation	CD9, CD81, CD63, CD82, CD47, MHC class I, syndecan, NT5E, CD59	LC-MS/MS	[48]
Bovine	Ultracentrifugation	FASN, XDH, BTN1A1, HSPA8, PLIN2, MFG-E8, IDH1, GDI2	LC-MS/MS	[61]
Bovine	Ultracentrifugation	CD63, CD81, TSG101, CD9	WB	[62]
Bovine	Ultracentrifugation	LGB, PAEP, CSN1S1	LC-MS/MS	[63]
Bovine (yak)	Ultracentrifugation	CD63, HSP70, TSG101	WB	[64]
Bovine	Density gradient centrifugation	CD63, HSP70, MFG-E8, BTN1A1	WB	[65]
Bovine	Density gradient centrifugation	Colostrum: B2M, Clusterin, PDGFC, CCN1 Mature milk: LTF, ANG1, LPO, QSOX1	LC-MS/MS	[57]
Horse	Ultracentrifugation	CD81, CD63, LGB, MFG-E8	MALDI-MS, MS/MS	[66]

### 4.2. RNA

Isolated fractions of EVs are enriched in various types of RNAs, including micro RNAs (miRNAs), as shown in Table 4 and Table 5. Additionally, Vesiclepedia contains a section on miRNAs in milk EVs. At present, by using combinations of methods to isolate EVs, the previously reported and often assumed strong association between the presence of (mi)RNAs and EVs is becoming less clear and may have been overestimated in many studies [67]. Whether this also holds true for milk needs to be studied, but again, differences between study outcomes suggests that the RNA composition also needs validation.

**Table 4 pharmaceuticals-14-01050-t004:** Main microRNAs identified in isolated extracellular vesicles from human milk.

Isolation Procedure	RNA	Detection	Study
Ultracentrifugation	hsa-miR-30d-5p, hsa-let-7b-5p, hsa-let-7a-5p, hsa-miR-125a-5p, hsa-miR-21–5p, hsa-miR-423–5p, hsa-let-7g-5p, hsa-let-7f-5p, hsa-miR-30a-5p, hsa-miR-146b-5p	RT-qPCR	[18]
Ultracentrifugation	Milk from mother who delivered pre-term infant: hsa-miR-22-3p, hsa-miR-148a-3p, hsa-miR-141-3p, hsa-miR-181a-5p, hsa-miR-320a, hsa-miR-378a-3p, hsa-miR-30d-5p, hsa-miR-30a-5p, hsa-miR-26a-5p, hsa-miR-191-5p; Milk from mother who delivered term infant: hsa-miR-22a-3p, hsa-miR-181a-5p, hsa-miR-148a-3p, hsa-miR-141a-3p, hsa-miR-30a-5p, hsa-miR-99b-5p, hsa-miR-191-5p, hsa-miR-378a-3p, hsa-miR-146b-5p, hsa-miR-30d-5p	UHTS	[68]
Ultracentrifugation	hsa-let-7c, hsa-miR-21, hsa-miR-34a, hsa-miR-146b, and hsa-miR-200b.	HTS	[69]
Density gradient centrifugation	hsa-miR-30d-5p, hsa-miR-148a-3p, hsa-miR-200a-3p, hsa-miR-200c-3p, hsa-let-7a-5p, hsa-miR-21-5p, hsa-let-7b-5p, hsa-let-7f-5p, hsa-miR-30a-5p, hsa-let-7g-5p	UHTS, RT-qPCR	[70]
ExoQuick-TC precipitation solution	hsa-miR-148a-3p, hsa-miR-22-3p, hsa-miR-30d-5p, hsa-let-7b-5p, hsa-miR-200a-3p	HTS	[71]
ExoQuick precipitation solution	hsa-miR-148a-3p, hsa-miR-30b-5p, hsa-let-7f-1-5p & -2-5p, hsa-miR-146b-5p, hsa-miR-29a-3p, hsa-let-7a-2-5p & -3-5p, hsa-miR-141-3p, hsa-miR-182-5p, hsa-miR-200a-3p, hsa-miR-378-3p	qPCR	[72]
ExoQuick-TC precipitation solution	hsa-miR-22-3p, hsa-miR-30d-5p, hsa-miR-148a-3p, hsa-miR-181a-5p, hsa-miR-141-3p, hsa-miR-30b-5p, hsa-miR-26a-5p, hsa-miR-92a-3p, hsa-miR-375, hsa-miR-30a-5p	UHTS	[73]
ExoQuick-TC precipitation solution	hsa-miR-148a-3p, hsa-miR-6073, hsa-miR-200c, hsa-miR-200b, hsa-miR-99a-5p, hsa-miR-30a-5p, hsa-miR-30d, hsa-miR-320-3p, hsa-let-7a-5p, hsa-miR-26a-5p	WGS	[74]
Total exosomes isolation reagent	miRNA from HIV-infected breast milk vs. uninfected; upregulated: hsa-miR-320e; hsa-miR-630; hsa-miR-148a-3p; hsa-miR-23a-3p; hsa-miR-378g; hsa-miR-30a-5p; hsa-miR-93-5p; hsa-miR-497-5p; hsa-miR-200b-3p; hsa-miR-200a-3p	NGS	[75]
ExoEasy maxi kit	hsa-miR-99b-3p, hsa-miR -96-5p, hsa-miR-550a-5p, hsa-miR-616-5p, hsa-miR-155-5p, hsa-miR-604	qPCR	[76]

**Table 5 pharmaceuticals-14-01050-t005:** Main microRNAs identified in isolated extracellular vesicles from livestock milk.

Milk Source	Isolation Procedure	RNA	Detection	Study
Bovine	Ultracentrifugation	bta-miR-223, bta-miR125b	RT-qPCR	[61]
Bovine	Ultracentrifugation	Colostrum: bta-miR-24, bta-miR-30d, bta-miR-93, bta-miR-106a, bta-miR-181a, bta-miR-200a, bta-miR451	RT-qPCR	[77]
Bovine	Ultracentrifugation	bta-miR-10b, bta-miR-143, bta-miR-10a, bta-miR-26a, bta-let-7a, bta-miR-21, bta-let-7f, bta-miR-222, bta-miR-99b, bta-let-7i	RT-qPCR	[78]
Bovine	Ultracentrifugation	bta-let-7a-5p, bta-let-7b, bta-let-7c, bta-let-7e and bta-let-7f, bta-miR-30a-5p, bta-miR-30d and bta-miR-30e-5p, bta-miR-148a, bta-miR-26a	HTS	[79]
Bovine	Ultracentrifugation, Density gradient centrifugation	bta-miR-223, bta-miR125b	RT-qPCR	[25]
Bovine	Total exosome isolation kit	bta-miR-26a, bta-miR-191, bta-miR-423-5p, bta-let-7f, bta-miR-30d, bta-let-7a-5p, bta-miR-27b, bta-let-7b, bta-miR-92a, bta-miR-125a	WGS	[80]
Bovine	ExoQuick reagent	- bta-miR-21, bta-miR-30a, bta-miR-92a, bta-miR-99a, bta-miR-223	qPCR	[81]
Porcine	Ultracentrifugation	ssc-miR-193a-3p, ssc-miR-423-5p, ssc-miR-320, ssc-miR-181a, ssc-miR-30a3p, ssc-miR-378, ssc-miR-191, ssc-let-7a, ssc-let-7f, ssc-let-7c.	HTS	[82]
Porcine	Density gradient centrifugation	ssc-let-7a-5p, ssc-miR-30a-5p, ssc-miR-191-5p, ssc-miR-21-5p, ssc-miR-30d-5p, ssc-let-7f-5p, ssc-let-7c, ssc-miR-200c-3p, ssc-let-7g-5p, ssc-miR-320a-3p	HTS	[70]
Porcine	ExoQuick exosome precipitation solution	ssc-miR-148a-3p, ssc-miR-182-5p, ssc-miR-200c-3p, ssc-miR-25-3p, ssc-miR-30a-5p, ssc-miR-30d-5p, ssc-miR-574-3p	HTS	[83]
Caprinae (sheep)	Total exosome isolation kit	oar-miR-26a, oar-miR-191, oar-let-7f, oar-let-7b, oar-miR-10b, oar-miR-148a, oar-let-7b, oar-let-7a, oar-miR-21, oar-let-7c	HTS, PCR	[84]

## 5. Functions of Extracellular Vesicles in Milk

Before introducing the reported functions of milk-derived EVs, which include coagulation, regulation of intestinal epithelium barrier function, anti-viral activity, and microbiome composition, it should be clear that such EVs can potentially have effects both on the mother and the infant. We will also give an overview of how the reported beneficial functions of milk-derived EVs have been explored, mostly in vitro and in animal models, for potential clinical utility, including osteoarthritis, cancer, and drug delivery.

### 5.1. Coagulation

Human milk triggers blood clotting [8], and recently, we demonstrated that the coagulant activity of human milk is due to the presence of tissue factor, which is present on EVs [33]. Tissue factor is a transmembrane protein. EVs are a suitable vehicle because they are surrounded by a phospholipid bilayer membrane, and because coagulation also requires a membrane surface [85]. To date, there is no evidence available that human milk contains coagulation factors other than TF.

Substantial nipple skin damage occurs in most breastfeeding women [86]. Fast activation of coagulation will quickly “seal the wound”, thereby reducing the risk of infection. Thus, one may speculate that tissue factor-exposing EVs in human milk may prevent infection and thus help to secure the mother’s health. In our study, surprisingly, bovine milk completely lacks this hemostasis-promoting property, and bovine milk-derived EVs are completely devoid of this tissue factor-coagulant activity, thus revealing a possibly functionally important difference between the milk of different mammalian species.

### 5.2. Intestinal Epithelium Barrier Function

The intestinal epithelial layer is essentially a barrier separating the body from bacteria that are present in the gastrointestinal tract. Regulation of tight junctions in this epithelial cell layer regulates the barrier permeability. When this barrier function becomes impaired, bacterial translocation can occur from the intestines into the blood, which leads to infection and sepsis [87].

The gastrointestinal tract of the pre-term infant is immature. After birth, nutrients such as oligosaccharides and immunoglobulins in milk facilitate the maturation of the gastrointestinal system [88]. The contribution of EVs to this barrier function has been studied in physiological and pathological models. With regards to the described effects of EVs to this barrier function, we can distinguish three main effects, i.e., the effects of EVs on proliferation, cell (tight) junctions, and mucin.

Firstly, in physiological models, proliferation of intestinal epithelial cells enhances the epithelial barrier function, and administration of milk-derived EVs promotes intestinal epithelial proliferation in rats [22] and pigs [89]. Furthermore, daily administration of porcine milk-derived EVs facilitated intestinal cell proliferation and intestinal tract development [85], with a possible involvement of the MAPK pathway [90].

In pathological (inflammatory) models, administration of human milk-derived EVs increased cell proliferation and decreased apoptosis in intestinal cells in vitro, and reduced intestinal epithelial injury and inflammation [91,92,93]. Additionally, rat and human milk-derived EVs promoted intestinal epithelial cell viability, enhanced proliferation of intestinal cells, and stimulated intestinal stem cell activity, thereby restoring the intestinal barrier function of newborns [22,23]. In a mouse colitis model, isolated EVs from bovine and human milk had an anti-inflammatory effect [94], whereas porcine milk EVs attenuated damage by promoting cell proliferation and tight junction formation via down-regulation of p53 [95]. Additionally, a diet with EV-depleted bovine milk induced intestinal inflammation in a mouse model [96]. Thus, the EVs present in milk seem to support intestinal integrity by inducing proliferation and regulation of permeability, but the underlying molecular mechanisms are unknown.

Secondly, the integrity of the intestinal epithelium depends on the junctions between cells [97]. There is evidence that milk EVs promotes the epithelial barrier function and reduces inflammation. Mice lacking a tight junction protein, kindlin-2, developed ulcerative colitis, which was reversed by treatment with bovine milk-derived EVs [98]. These findings suggest that milk EVs affect the tight junction function, and thus are involved in regulation of the permeability of the intestinal barrier.

Thirdly, the intestinal epithelium is covered by a layer of mucin, a protective additional barrier that is produced by goblet cells. Loss of mucin and reduced numbers of goblet cells are both associated with disease development of intestinal inflammation [99]. In an inflammation mucosal injury model, human milk-derived EVs increased the mucus production [93,99]. Treatment with bovine milk-derived EVs induced an increased expression of mucin-2; similarly, the addition of bovine milk-derived EVs to goblet cells stimulated mucin production and secretion, and prevented the development of intestinal inflammation [60]. In addition to this in vivo model of mucosal injury, intestinal mucosal injury and inflammation were attenuated by the administration of human milk EVs [93,99].

Although all evidence thus far confirms that milk-derived EVs promote epithelial barrier function, one must keep in mind that most, if not all, of these studies have used impure EVs and experiments were performed in mixed-model species, which hampers the interpretation of the reported results.

### 5.3. Anti-Viral Activity

Milk-derived EVs affect the immune responses of the mother by regulating the activity of immune cells [58,100]. For example, milk-derived EVs promoted the uptake of human immunodeficiency virus (HIV)-1 by macrophages and inhibited uptake by T cells [21,26]. Binding of EVs to antigen-presenting cells inhibited HIV infection of both dendritic cells and CD4+ T cells, which in turn may explain why an HIV-infected mother does not infect a breast-feeding baby [101]. Such an anti-viral activity was also reported for cytomegalovirus (CMV). CMV is regularly transmitted from the mother’s lactation epithelium to infants. Although transmitted CMV can be of great risk to the preterm infant, the reported cases are rare, suggesting that there are one or more protective mechanisms present in breast milk. EVs isolated from colostrum impaired the attachment of CMV to human foreskin fibroblast-1 cells, which was abolished by trypsin pretreatment of EVs, indicating a possible involvement of EV surface proteins [102].

### 5.4. Microbiome

The gastrointestinal epithelium forms a barrier, and thus prevents the entry of bacteria and bacterial toxins into the intestinal mucosa, and, eventually, into the systemic circulation [103]. This barrier function is dynamic and regulated by multiple stimuli [104]. Balanced microbial communities in the intestinal tract itself also support this barrier function. Bovine milk-derived EVs facilitated the gut microbiota composition, modulated bacterial metabolites, and regulated local intestinal immunity. For example, loss of Lachnospiraceae, a family of anaerobic, spore-forming bacteria, is associated with development of intestinal bowel disease [105,106], and the presence of Lachnospiraceae is supported by EV-enriched diets in mice [107]. Thus, potentially, milk EVs may be a valuable nutritional probiotic for human infants.

## 6. Therapeutic Application of Milk-Derived Extracellular Vesicles

### 6.1. Osteoporosis

Osteoporosis is a systemic skeletal disease characterized by weak and brittle bones, resulting in bone fragility and fracture susceptibility. EVs from bovine milk were reported to have both direct and indirect effects on bone mineral density. Regarding the direct effect, the addition of bovine milk-derived EVs regulated osteoblast and osteoclast differentiation and proliferation in vitro [108,109,110]. However, because milk-derived EVs will be present within the gastrointestinal tract, this direct effect seems unlikely to occur in vivo, because intact EVs would need to pass the epithelial barrier, travel to bones via the blood, and then regulate osteoblast and osteoclast differentiation and proliferation. Regarding the indirect effect, mice suffering from osteoporosis receiving bovine colostrum-derived EVs had higher bone mineral density compared to control mice. Because these changes were paralleled by changes in the microbiome, these authors speculated that bone mineral density may be affected by (indirect) EV-induced changes to the microbiome [111].

### 6.2. Arthritis

Consumption of bovine milk-derived EVs by mice suffering from arthritis protected cartilage and reduced inflammation of the bone marrow [112]. Interestingly, this protective effect of milk-derived EVs was paralleled by changes in the microbiome, again supporting the observation that milk-derived EVs may be part of the “gut-bone axis” [113,114].

### 6.3. Cancer

Oral administration of camel milk, or of camel milk-derived EVs orally or by local injection, reduced breast tumor progression in mice by inducing apoptosis of cancer tissue, increasing antioxidant enzyme activity, decreasing lipid peroxidation, and increasing oxidative stress in tumor tissue compared to the control group [115]. In contrast, human milk-derived EVs enhanced the epithelial-mesenchymal transition of breast cancer in vitro [116]. More work needs to be done to confirm how purified milk-derived EVs affect tumor growth and metastasis.

### 6.4. Drug Delivery

Studies are ongoing to investigate whether milk-derived EVs can be used as a cheap drug delivery vehicle instead of liposomes [117,118]. Most of these studies, however, suffer from multiple shortcomings. Firstly, milk EVs are difficult to isolate, and because there are no standard operating procedures, contamination with milk fat globules, casein micelles and soluble components cannot be excluded. Secondly, milk contains bacteria and viruses, and thus milk is not a sterile body fluid. Although there are procedures to disinfect milk, these procedures may also damage EVs. Especially for intravenous applications, sterilization of milk seems essential, whereas for oral applications it is recommended. Thirdly, milk EVs from non-human mammals other than cows and possibly relatives may expose tissue factor and thus trigger blood clotting. If so, intravenous administration of such EVs would directly trigger thrombus formation, which may even be lethal. Due to these limitations, administration of milk-derived EVs seems unsafe and requires additional studies.

The idea behind using (milk-derived) EVs for drug administration is to lower the dose of toxic anti-cancer drugs by improving their bioavailability. Thus, EVs are used as lipid carriers instead of liposomes, and by loading toxic drugs into milk-derived EVs, the total dose can be lowered [119]. For example, the therapeutic effectivity of paclitaxel, cisplatin, and doxorubicin was enhanced by packaging these drugs into EVs [120,121], and nude mice bearing human lung tumors were treated successfully with intravenously administered, paclitaxel-loaded bovine milk-derived EVs with low systemic and immunologic toxicities [122]. Whereas paclitaxel, cisplatin and doxorubicin are hydrophilic, milk-derived EVs have also been used as carriers of hydrophobic compounds such as curcumin, a substance isolated from the root of the turmeric plant reported to have anti-cancer potential [123]. When curcumin is incorporated into EVs from buffalo milk, curcumin had improved bioavailability compared to free curcumin [124]. Once loaded into EVs, curcumin was more easily taken up by cells [125], and tumor growth inhibition was stronger than non-packaged curcumin [120]. Finally, milk-derived EVs have also been used as carriers of siRNAs [126]. Taken together, milk-derived EVs may have potential for drug delivery, but one has to be careful because milk is not sterile and EVs may trigger blood clotting. EVs from mesenchymal progenitor cells or other mesenchymal cell types may be better for intravenous administration [127,128].

## 7. Survival of Extracellular Vesicles from Mothers’ Milk in the Infants’ Digestive Tract

Milk directly enters the gastrointestinal tract of the infant and thus faces digestion and degradation. Digestion affects the presence of bioactive components, including those associated with EVs. Thus, the question is whether EVs or the bioactive components of EVs may survive under physiological conditions of oral administration.

Digestion encompasses food entering the mouth, stomach, small intestines and colon. During digestion, food turns into macromolecules, which are absorbed as nutrients by the blood, and waste. In the oral digestion phase, milk is mixed with a minor volume of saliva. Then, in the gastric digestion phase (stomach), the EVs encounter an acidic environment, which is followed by enzymatic degradation in the small intestines, etc. Infants, especially pre-term infants, have an immature digestive system that is characterized by less harsh conditions than those encountered in the mature digestive system. For example, the pH of the gastric fluid of infants is higher than in adults [68,129,130], and concentrations of degrading enzymes, such as amylase, trypsin and lipase [131], and bile salts [130] are lower. In vitro, milk-derived EVs survive simulated gastric digestion and/or simulated intestinal digestion [73,130,132], which supports data indicating that endogenous milk EVs are beneficial, at least in infants [25,68,73,133].

## 8. Possible Uptake of Milk-Derived Extracellular Vesicles

As explained, drinking milk brings EVs into the infant’s digestive tract, and such EVs are likely to survive the relatively mild environmental conditions. To be taken up by the intestinal epithelial cells, the first barrier milk EVs have to pass is a layer of mucus covering the intestinal epithelium. Whether milk EVs are trapped in this layer, or how they can move across this layer, is unknown. In vitro studies have claimed that EVs bind to and are taken up by intestinal epithelial cells [73,112,134], vascular endothelial cells [135], macrophages [44], and splenocytes [112], and in a mouse model, milk-derived EVs were shown to be absorbed by the gut and to accumulate in several organs [136]. However, whether uptake of EVs by epithelial cells is affected by a layer of mucus in vitro is obscure. Furthermore, in the mouse model, EVs were labeled with a lipid membrane dye, and using such dyes is more often than not a source of artefacts [137].

## 9. Summary

Milk is an important body fluid as it is both consumable and the recommended food for newborns, since milk reduces intestinal complications. There is compelling evidence that milk contains EVs. At least some of the beneficial effects of milk may be attributed to the presence of EVs (Figure 2). Due to the lack of proper methods to purify EVs from milk, most results thus far regarding their biochemical composition and function awaits confirmation.

## 10. Outlook

The recommended practice of exclusively breastfeeding to at least 6 months after birth is not applied to all infants [138], and the benefits of human milk compared to formula encourages the current efforts of establishing donor milk banks for wider applications of human milk feeding to infants. To establish such biobanks, donor milk must be collected, transported, and stored. Because milk contains potential pathogens, collected donor milk needs to be disinfected. Although several disinfection methods are available, including pasteurization, at present it is unknown to which extent such methods affect the functions of milk and endogenous EVs. One may hypothesize that the coagulant activity of milk, which can be solely attributed to EVs present in milk, can be used as a read-out to evaluate the effects of collection and handling, including disinfection, and at the same time may provide insight into the intactness and functionality of at least a fraction of EVs present in such milk samples.

## Figures and Tables

**Figure 1 pharmaceuticals-14-01050-f001:**
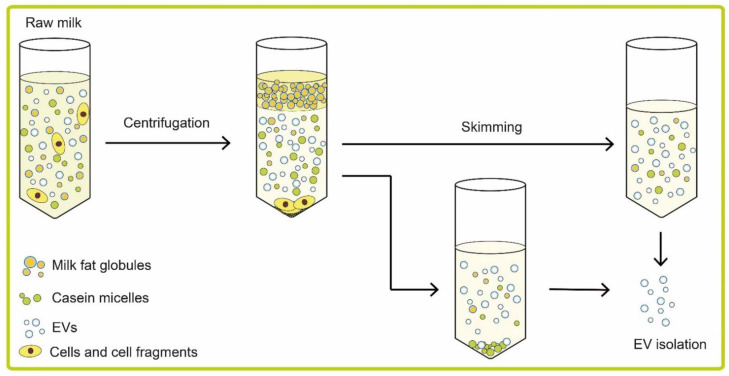
Processing of milk before isolation of extracellular vesicles. Raw milk contains cells, cell fragments, milk fat globules, casein micelles, and extracellular vesicles (EVs). As a first step, cells, cell fragments and milk fat globules are mostly removed by centrifugation. Milk fat globules will float to the top due to their low density, and cells and cell fragments will form a pellet. The middle layer contains casein micelles and EVs. This middle layer is the starting material for the isolation of EVs. Casein micelles can be removed by acid-induced or enzyme-induced precipitation, centrifugation, or by EDTA-induced dissociation.

**Figure 2 pharmaceuticals-14-01050-f002:**
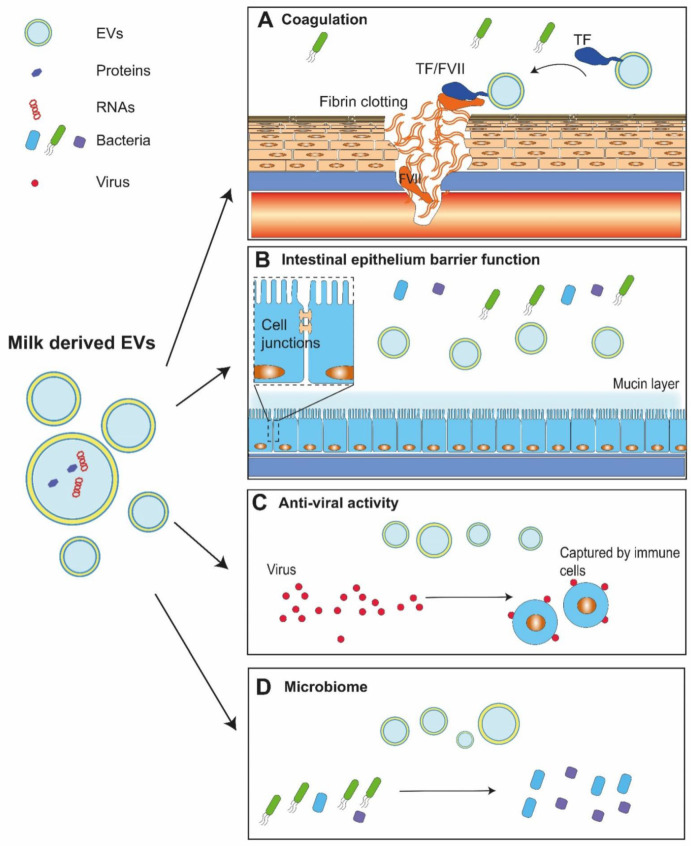
Functions of milk-derived extracellular vesicles. Milk contains extracellular vesicles (EVs). (**A**) EVs in human milk expose tissue factor (TF), the protein that triggers blood clotting by binding and activating factor VII (FVII), thereby promoting haemostasis and wound healing. (**B**) Milk EVs may survive the gastrointestinal conditions of newborns, and such EVs are thought to participate in immune responses and homeostasis of the gastrointestinal tract. (**C**) Milk EVs attenuate virus infection with help from immune cells. (**D**) Milk EVs may kill harmful bacteria and promote the growth of beneficial bacteria within the gastrointestinal tract, which may have beneficial effects on growth and development of the newborn.

**Table 1 pharmaceuticals-14-01050-t001:** Particles present in milk.

		Extracellular Vesicles	Casein Micelles	Milk Fat Globules
		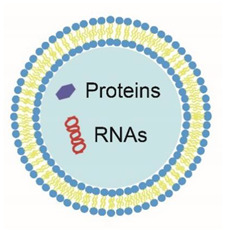	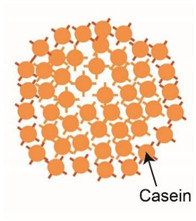	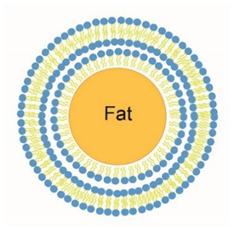
Membrane structure		Phospholipid bilayer	No membrane	Phospholipid trilayer
Diameter	Human	50 nm–265 nm [21,22,23]	30 nm–75 nm [29]	200 nm–11 μm [30]
	Cow	44 nm–200 nm [24,25]	20 nm–600 nm [31]	200 nm–15 μm [32]

## Data Availability

Data sharing not applicable.

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
