# Peer review of "Extracellular Vesicles in Human Milk"

_pharmaceuticals, 2021, doi:10.3390/ph14101050_

Round 1

Reviewer 1 Report

in my view the work is substantially improved

Author Response

Reviewer 1, Comments and Suggestions for Authors

In my view the work is substantially improved.

We thank the reviewer for her or his comments.

Reviewer 2 Report

I have been repeatedly through the revised version. I am afraid that I can't identify major differences regarding the improvement of the manuscript.
The authors though did introduce a major difference in the title narrowing their scope on milk from human origin. However, there is much information of comparative nature through the whole text, from experimental procedures/outcomes to the justification of milk EVs biological activities. In addition, coagulation remains in focus.

In addition to the previous comments, I have multiple considerations, because milk contains cellular and non-cellular material. So far, EVs isolation cannot warrant the exclusion of the rest of "other" bioactive material. In addition, it is not established how chelating agents may interfere with milk EVs active components.
The presence of typical EVs markers it is not discussed. There has been a selection of proteins (among them tetraspanins may be used as positive plasma membrane shedding markers) and RNAs: i) it is not understood how this selection was made – is there any cut-off?, ii) it is not understood how it adds to the multiple markers proposed by the guidelines, iii) it is not understood how it will help us understand their role in potential milk biological activity.
Moreover, it still appears that there is no good understanding of mammary gland physiology, potential differences between species, and the risk of extrapolation of in vitro experiments. In this respect, it is quite premature not to discuss possible (pro)oncogenic effects. In addition, milk proteome shows inter-species consistencies and differences depending on protein function [https://proteomesci.biomedcentral.com/articles/10.1186/s12953-016-0110-0], and it is hard to understand how this may or may not reflect or counteract the differences in EVs content. It is also not to be excluded that various cells may shed material depending on their functional state, and this material may just be a by-product of cellular physiology.

In summary, I believe that the manuscript has many weak points regarding structure and interpretation. It is not clear how the Pharmaceuticals’ reader would benefit from the “composition” or the “function” sections, and if there would be separate conclusions. It is also unclear whether the manuscript would guide someone investigating milk EVs against their “natural” targets or as experimental drug. and whether it is premature to proceed to such attempts, e.g. EVs loading, when their natural loading hasn’t been identified yet. The authors finally could apply some constructive criticism while enumerating the studies.

Reviewer 3 Report

This review focus on why milk contains extracellular vesicles (EVs) and what their function could be. The group demonstrated that human milk-derived EVs promote blood coagulation, which is not the case for bovine EVs suggesting differences in the biological functions of milk-derived EVs between species. The authors summarize the current knowledge regarding the presence and biochemical composition of milk EVs, their function(s) and potential clinical applications such as probiotics, and problems that milk EVs encounter in in vivo settings.

A mostly well written manuscript which is of interest in the EV field. Not many reviews are published up to now focussing on milk derived EVs and their effects and functions.

General points

  • The authors have listed proteins and miRs, which are identified in milk EVs. What is known about lipids/fatty acids? Aren’t there typical lipids/fatty acids, which are carried in the milk EVs?
  • In the section 6.1 with the title “Osteoarthritis” two musculoskeletal disorders are mixed and confused: osteoarthritis and osteoporosis. I suggest to separate this part in 2 subchapters “Osteoarthritis” and “Osteoporosis” and describe therapeutic applications of milk-derived  EVs for both disorders separately.
  • 3. Drug delivery: Here it should be made clear that EVs from other sources than milk might be better suited as drug delivery vehicles than milk EVs. This goes in particular for EVs derived from mesenchymal progenitor cells or other mesenchymal cell types. However, the advantages of milk derived EVs over those from cells should be also discussed.

Specific points

Abstract: please add “human” to the sentence “milk-derived EVs promote coagulation”.

6.3. Drug delivery: page 11. Lane 357: please add “free” to “compared to curcumin”.

Tables

Table1: Please write “no membrane” instead of simply “No”.

Tables 4 and 5: Please change “RNAs” in the title to “micro RNAs” as only those are listed.

Round 2

Reviewer 2 Report

I would like to thank the authors for providing an extensive critical commentary. Honestly, I would prefer this kind of approach to be followed upon manuscript building, even with some hints about "personnal communication". My point is that the field is suffering from concrete evidence, and the concurrent and extensive investigation of milk components in animals doesn't mitigate the confusion.

Given the current MISEV guidelines, one would easily go through the tables to repeat or extext the list of components, rather than initiating a more structured inquiry. Tissue/organ physiology would be more helpful in putting everything in context, and I agree that although platelets may appear a disparate system, mammary gland microbleedings and its changes relevant to breastfeeding may guide a new route of investigation.

Field being still fuzzy, and authors willingness to comment on these issues, I would expect the Editor now to make a critical decision.

-My final option for revisions is indicative-

Wish you good luck with your future endeavours!

Reviewer 3 Report

The authors have mostly responded fine to my comments. However, following comment was insufficiently answered: 

In the section 6.1 with the title “Osteoarthritis” two musculoskeletal disorders are mixed and confused: osteoarthritis and osteoporosis. I suggest to separate this part in 2 subchapters “Osteoarthritis” and “Osteoporosis” and describe therapeutic applications of milk-derived  EVs for both disorders separately.

The authors have not separated both disorders into 2 subchapters as recommended but only changed the title. This is not sufficient as the chapter is as confusing as before. A substantial revision of this chapter is required

Author Response

This manuscript is a resubmission of an earlier submission. The following is a list of the peer review reports and author responses from that submission.

Round 1

Reviewer 1 Report

The authors submitted a review article entitled “Extracellular vesicles in milk”. The field is growing, and it would be of interest to the scientific community. However, there are several issues that should be addressed in order to cover the field and provide background information on the i) preventive or ii) therapeutic activity of milk content to the readers of Pharmaceuticals Journal.

There are major issues regarding extracellular vesicles description. Please include the [Minimal information for studies of extracellular vesicles 2018 (MISEV2018): a position statement of the International Society for Extracellular Vesicles and update of the MISEV2014 guidelines], and revise all passages accordingly.

At the technical level, and to improve the organization of the manuscript, the authors could also advise [https://cdn.ymaws.com/www.isev.org/resource/resmgr/misev2018_checklist.pdf]

Unclear meaning in L144 “the total biochemical mass”, L150 “genetic information”. Please revise this section and keep it minimal.

Regarding the scope of the manuscript, it is not clear what is the focus of the manuscript regarding milk origin. There is a confusion whether the authors are intended to support the benefits of breast feeding in humans, or provide evidence about milk in general.

In any case, the authors rely on a small number of relevant publications. I believe that there are several publications on different lactating species, humans included [https://www.frontiersin.org/articles/10.3389/fnut.2020.00022/full].

It would be also necessary to have a brief introductory part of milk nutrient content, and species differences, as there is an extensive mention to them later.

It would be also essential to underscore initially fat milk content. (References to colostrum may or may not be necessary to be included at this point.) The reference could be also discussed in the context of fat removal and possible interference with EVs during isolation.

Regarding EVs content, it is understood that isolation methods may have a serious impact on findings. However, I found hard to confirm the Exocarta/Vesiclepedia claims. Another point is that the authors should provide a more plausible presentation of protein content vs method/provenance. It is neither understood how the authors came up with a very short list of proteins for Tables 2 & 3, or why there is no evidence of protein content for studies included in Tables 4 & 5. There is also a mix-up regarding literature screening and the necessity to rely on these databanks. As long as other sources are available, the authors could retrieve up-to-date data, and finally conclude and provide constructive criticism on the methods, and later on the repositories.

The third major issue is relevant to the possible preventive/therapeutic role of milk content, and to Pharmaceuticals Journal scope. First, it is essential to discuss intra-species events. This fact is marginally introduced - the lack of context is also a major problem in the majority of EVs studies. Therefore, it would be better for the authors to start with GI physiology and development and discuss and elaborate the issues presented in chapter 7-8. Then, they could discuss potential effects of any milk content, starting from early age.

There is also a trend within the manuscript to underscore coagulation, and a need to put this information into the right context. Please, consider that there are other factors implicated in the cascade. In addition, TF physiology and experimental detection in humans has been described extensively [https://www.ahajournals.org/doi/full/10.1161/atvbaha.117.309846]. Such difficulties may also appear when studying EVs. Furthermore, milk is expected to contain at least EVs originating from the mammary gland. This matter is not addressed at all, and I believe that breast physiology during lactation (and lactation stage) may have a serious impact on EVs content. Older articles in the field could be of use, indicative references [https://breast-cancer-research.biomedcentral.com/articles/10.1186/bcr1871 ; https://pubmed.ncbi.nlm.nih.gov/18032632/].

Finally, the references to milk contaminants, and milk supply chain, should be also revised, as there may be evidence of poor hygiene or opportunistic infection in the mother. I also believe that there are other references regarding anti-microbial milk peptides and simple nutrients that could circumvent infection initiation, and facilitate microflora introduction and restauration in the newborn.

Regarding language and format, many problems may be fixed upon careful literature study, and content reorganization. Reference should be also reformatted and updated.

Reviewer 2 Report

In the work, „Extracellular vesicles in milk“, the authors Hu et al. present a nice and insightful review on a particular subject that has only recently gained substantial attention. The work is well written and concise, however, some substantial aspects should be improved.

In particular, it is a collection of observations, but some more discussion on the relevance / reliance of some findings would help.

Some information is lacking.

What is the actual content (percent of volume) of EV in milk?

Do the EV carry surface molecules at the outer membrane that interact with receptors – in addition to the described interaction with tissue factor?

How do they actually interact with cells, e.g. macrophages (by which binding mechanisms?)

Is the membrane of EV just a phospholipid bilayer or does it contain proteins?

Some paragraphs of the work contain just some observations, but the scientific content is not very coherent, e.g. in the paragraph on cancer. Whether camel milk really is protective against cancer appears a little questionable to me; such quotations might be a bit discussed.

Minor points:

p1 line 35

 heriditary bleeding  - write hereditary

p1 line 44

 ecotosomes – write ectosomes

p 11 line 375

new born – write newborns